# Skipping Exon-v6 from CD44v6-Containing Isoforms Influences Chemotherapy Response and Self-Renewal Capacity of Gastric Cancer Cells

**DOI:** 10.3390/cancers12092378

**Published:** 2020-08-22

**Authors:** Silvana Lobo, Carla Pereira, Carla Oliveira, Gabriela M. Almeida

**Affiliations:** 1i3s–Instituto de Investigação e Inovação em Saúde, 4200-135 Porto, Portugal; slobo@ipatimup.pt (S.L.); carlap@ipatimup.pt (C.P.); carlaol@i3s.up.pt (C.O.); 2IPATIMUP–Institute of Molecular Pathology and Immunology of the University of Porto, 4200-135 Porto, Portugal; 3Doctoral Programme in Biomedicine, Faculty of Medicine of the University of Porto, 4200-319 Porto, Portugal; 4FMUP–Faculty of Medicine, University of Porto, 4200-319 Porto, Portugal

**Keywords:** CRISPR/Cas9, chemoresistance, exon skipping, morpholinos, cell survival, stomach neoplasms

## Abstract

De novo expressed CD44 isoforms containing exon-v6 are frequently associated with gastric cancer (GC) aggressiveness, and may predict chemotherapy response in vitro. Whether exon-v6 itself is responsible for conferring these properties to CD44v6-containing isoforms remains to be elucidated. CRISPR/Cas9 and Phosphorodiamidate Morpholino oligomers (PMOs) were used to induce specific exon-v6 skipping, maintaining the CD44 reading frame, in two GC cell lines endogenously expressing CD44v6. Cisplatin and 5-fluorouracil treatment response, and self-renewal ability was compared between CRISPR/Cas9-edited, CD44v6 knockdown and mock cells. We obtained homozygous genome-edited cell lines with exon-v6 deletion. Edited cells transcribed CD44v isoforms presenting in frame v5–v7 splicing, mimicking exon-v6 skipping. Results showed that removing specifically exon-v6 sensitizes cells to cisplatin and impairs cells’ self-renewal ability, similarly to CD44v6 knockdown. In parallel, we also tested a clinically feasible approach for transient exon-v6 skipping with a PMO-based strategy. We demonstrate that exon-v6 specific removal from CD44v isoforms increases cell sensitivity to cisplatin and impairs GC cells self-renewal. We trust that a PMO approach designed towards CD44v6 overexpressing GC cells may be a suitable approach to sensitize tumor cells for conventional therapy.

## 1. Introduction

Gastric cancer (GC), the third leading cause of cancer-related death worldwide, is often diagnosed in advanced/unresectable stages, where patients are treated with conventional chemotherapy and have an expected survival of only 1 year [1]. Thus, it is crucial to improve treatment effectiveness and patients’ outcome. CD44 is a family of glycoproteins encoded by CD44 gene (locus 11p13) [2,3], that mediate cell-cell and cell-matrix contact essential for tissue integrity and maintenance. Human CD44 gene (NG_008937) is composed of two terminal constitutively expressed portions (exons 1–5/16–18/20), and in between, by different combinations of alternatively spliced exons (exons 7–15/v2–v10) [2,4,5]. CD44v6-containing isoforms encompass a group of isoforms, including variant exon-6 (v6) [3,6], a cancer stem cell (CSC) marker widely associated with poorer patient prognosis, increased invasion, metastization and drug resistance in several cancers, including GC [7,8,9,10,11,12]. Blockade of CD44v6 delays tumor growth and metastization in several in vivo models of pancreatic [13] and colorectal cancer [14]. Therefore, targeting CD44v6-expressing cancer cells arises as a suitable cancer therapy candidate. This strategy was tested in head and neck squamous cell carcinoma with the development of a monoclonal antibody against CD44v6 (BIWA) [15], combined with mertansine, a chemotherapeutic cytotoxic drug. This clinical trial was discontinued, as one participant passed away due to severe skin toxicity, since keratinocytes express CD44v6-containing isoforms [16]. CD44v6-containing isoforms are not expressed in normal gastric mucosa, but become overexpressed in pre-neoplastic gastric lesions and tumors [17]. Recently, we reported that GC cell lines expressing CD44v6-containing isoforms are more resistant to cisplatin than cells lacking CD44v6 expression [18]. Still, it remains to be elucidated whether it is exon-v6 itself, or the isoforms containing exon-v6, that modulate chemotherapy response in cancer cells. Herein, we hypothesize that the presence/absence of exon-v6 in CD44v6-containg isoforms conditions the chemotherapy response of GC cells, and that specific exon-v6 skipping can sensitize GC cells to chemotherapy.

In this study, we used CRISPR/Cas9 to remove exon-v6 from the CD44 locus, by targeting splice-sites and/or adjacent regions, to generate specific exon-v6 skipping from CD44v6-containing isoforms in two GC cells lines. Our results show that specific exon-v6 skipping induces increased cell sensitivity to cisplatin and decreased self-renewal capacity, to the same extent as the full depletion of CD44v6-containing isoforms. In sum, our data highlight a feasible alternative to full CD44v6-containing isoforms depletion, raising the possibility of using exon-v6 skipping as targeted therapy to delay GC cells self-renewal and sensitize gastric tumors to chemotherapy. This is expected to decrease off-target effects observed in previous approaches.

## 2. Results

We aimed at understanding if the presence of exon-v6 in CD44v isoforms may affect GC chemotherapy response and cell self-renewal, and for this, we developed an exon-v6 specific skipping strategy.

### 2.1. CRISPR/Cas9 Editing at Exon-v6 Splice-Sites Generates CD44v Isoforms Lacking Exon-v6

We established a CRISPR/Cas9 strategy to target the adjacent areas of exon-v6 (Figure 1A), by designing single guide RNAs (sgRNAs) (g1–g6) in the exon-v6 vicinity (Figure 1B,C), to achieve exon-v6 skipping. After genome editing, we obtained two edited cell lines for MKN45 (O26 and O15) and GP202 (O26 and O25). To evaluate whether our strategy successfully removed exon-v6 from CD44v isoforms, genomic DNA (gDNA) was amplified, using primers for exon-v6 flanking introns (Primers C/F, Table 1). Resulting products were smaller in edited cells than in wild-type (WT) and mock controls (Figure 2A, left panel). This was confirmed by Sanger sequencing (Figure 2B, panel 1–4), showing that all CRISPR/Cas9 edited cell lines presented homozygous deletions encompassing exon-v6 (Figure 2C). To evaluate specific exon-v6 skipping, complementary DNA (cDNA) was amplified with primers in v5 and v7 adjacent exons (Primers B/E, Table 1). All edited cell lines presented a smaller transcript, corresponding to splicing between exons v5-v7 and in-frame exon-v6 skipping (Figure 2A, right panel; Figure 2B, panels 5–6). By real-time quantitative PCR (RT-qPCR), we verified that our editing strategy was specific for CD44v6-containing isoforms, as no exon-v6 expression was detected in edited cells, while total CD44 expression was maintained (Figure 3A). By flow cytometry, we showed that edited cell lines presented depletion of the exon-v6 encoded peptide, while maintaining equivalent total CD44 protein levels as mock cells (Figure 3B). To demonstrate that only the v6-encoded domain was skipped, we performed co-immunofluorescence for CD44v6 and CD44v9, a variant exon downstream of exon-v6 that is often included in CD44v6-containing isoforms. As expected, exon-v6 and exon-v9 were co-detected in mock cells, while in edited cells, only exon-v9 was detected (Figure 3C). When using v6-directed siRNAs (CD44v6 KD cells), both exon-v6 and exon-v9 expression disappeared, demonstrating that exon-v6 and exon-v9 are both present in the same CD44v6-containing isoforms detected with this approach. These data support that, exon-v6 skipping in edited cells occurs without interfering with downstream exon-v9, demonstrating that CD44v isoforms expression remains unchanged.

### 2.2. CD44v6-Containing Isoforms and Exon-v6 Itself Are Positive Modulators of Cisplatin Response

We previously demonstrated that MKN45 and GP202 GC cell lines, which express CD44v6-containing isoforms, are less sensitive to cisplatin than CD44v6-depleted counterparts [18]. To evaluate if exon-v6 is, itself, responsible for modulation of chemotherapy response in these cell lines, we performed short and long-term assays in the presence of cisplatin and 5-fluorouracil (5-FU). For that, we compared mock cells treated with scramble siRNA (scramble), with CD44v6 knockdown (KD) cells, or with exon-v6 edited cells. In both cell lines, 48 h after treatment, we observed that CD44v6 KD cells survived better than scramble in presto blue (PB) and sulforhodamine B (SRB) assays (Figure 4, left panel). The outcome of short-term treatment in MKN45 and GP202 exon-v6 edited cells mimicked the observations for CD44v6 KD for at least one clone per cell line, but only for cisplatin treatments, and mainly for the PB assay. For SRB assay, only MKN45_O26, treated either with cisplatin or 5-FU, mimics the significant cell survival increase observed in CD44v6 KD. Regarding the long-term survival, clonogenic assay demonstrated the opposite, as CD44v6 KD cell survival significantly decreased when compared to scramble, in response to both drugs (Figure 4, left panel). The same was true for both GP202 edited cells, which presented a statistically significant or strong tendency for decreased cell survival to cisplatin compared to control (Figure 4, bottom right panel). Overall, these results demonstrate that exon-v6 skipping mimics to some extent the effect that CD44v6 KD have in GC cell survival. Moreover, it consolidates the role of both CD44v6-containing isoforms, and exon-v6 alone, as positive modulators of GC survival.

### 2.3. Exon-v6 Depletion Decreases Self-Renewal Capacity of GC Cells to the Same Extent as Knocking-Down CD44v6-Containing Isoforms

Next, we tried to understand whether self-renewal capacity, which can be assessed by calculating basal colony formation efficiency, was different between mock, CD44v6 KD and edited cells. We observed that both CD44v6 KD cell lines presented extremely low colony formation efficiency compared to scramble (Figure 5). Interestingly, the two edited clones that behaved similarly to CD44v6 KD (MNK45_O26 and GP2020_O25), in short- and/or long-term assays with cisplatin and 5-FU, were also those presenting a statistically significant decrease in colony formation efficiency (Figure 5). We tried to understand if deletion size or break-points’ location could explain the different results obtained for the edited clones. Most likely, this does not provide an explanation, as MKN45_O26 and GP202_O26 share exactly the same deletion (Figure 2C), and behave differently regarding drug response and colony formation efficiency. In summary, the latter results clearly demonstrate that exon-v6 skipping can have the same effect on cell self-renewal as full KD of CD44v6-containing isoforms; however, in the future, it would be important to extend this analysis to additional GC cell lines.

### 2.4. Transient Exon-v6 Skipping Using Phosphorodiamidate Morpholino Oligomers (PMOs)

Our results suggest that specific exon-v6 skipping significantly decreases GC cells self-renewal, while sensitizing GC cells to cisplatin in long-term assessment. Therefore, we hypothesize that using exon-v6 skipping therapy in CD44v6 expressing gastric tumors could improve GC patients’ response to chemotherapy. Hence, we tried to use a morpholino-based transient strategy, which is already approved in the clinics for other diseases, to perform CD44 exon-v6 skipping. We designed PMOs directed to exon-v6 flanking splice-sites (Figure 6A,B). PMOs were expected to mask splice-sites, leading to the production of CD44v transcripts lacking exon-v6. We successfully reproduced the exon-v6 CRISPR/Cas9 editing in GP202 cells, either by using a single PMO or combining two PMOs. RNA analysis revealed the production of a smaller transcript (Figure 6C) compared to the one observed in parental cells, of which sequencing was confirmed to represent exon-v6 skipping (Figure 6D). Using immunofluorescence, we demonstrated that exon-v6 is undetectable, as a result of exon skipping, whereas total CD44 remains expressed (Figure 6E). These results indicate that the use of at least one PMO targeting an exon-v6 splice-site efficiently induces exon-v6 splicing out, creating an opportunity for clinical intervention.

## 3. Discussion

Aiming to unveil whether exon-v6 is, by itself, responsible for increased chemotherapy resistance in CD44v6 expressing GC cells and cell self-renewal, we generated permanent exon-v6 skipping models. We successfully obtained four edited cell lines with specific exon-v6 skipping, where in-frame splicing between exons v5 and v7 occurred, and where CD44v6 protein expression was knocked-out, while total CD44 expression was maintained. This work confirms that CRISPR/Cas9 editing is efficient at generating CD44v isoforms lacking exon-v6. Moreover, our work indicates that this can be useful to develop in vitro models to study the function of specific exons, by inducing targeted exon skipping without destroying the protein of interest.

In order to mimic patients’ treatments, we performed short-term drug assays, mimicking the acute phase after chemotherapy, and long-term drug assays, mimicking the response after a chemotherapy cycle. Short-term analysis suggests that, with CD44v6 KD, cells survive better to cisplatin and 5-FU, compared to CD44v6 expressing cells. CD44v6 KD is associated with several changes in tumor matrix organization [19], and, therefore, we hypothesize that, upon CD44v6 KD, cells activate specific signaling pathways to compensate CD44v6 loss, leading to increased survival right after treatment. However, further studies would be necessary to confirm this hypothesis. Nevertheless, this effect seems transient, since, in the more biologically relevant long-term analysis, CD44v6 KD decreases cell survival, compared to CD44v6 expressing cells. Regarding exon-v6 depleted cells, two edited clones, MKN45_O26 and GP202_O25, present a similar behavior to that of CD44v6 KD in short-term cisplatin treatment. MKN45_O26 also presents the same behavior in response to 5-FU. Importantly, in long-term assays, GP202 edited cell lines respond similarly to the KD of CD44v6 isoforms in response to cisplatin. Therefore, we believe that specific exon-v6 depletion can sensitize GC cells to cisplatin, which may be dependent on the cellular context. Although all edited cell lines present specific exon-v6 skipping, they do not all show similar results in chemotherapy response. This might be caused by the different DNA breakpoints. For instance, MKN45_O15 still possesses half of the exon-v6 sequence while MKN45_O26 has complete deletion. Although both GP202 edited clones present complete exon-v6 deletion, GP202_O26 retains more 46 bp than GP202_O25 in the adjacent intron. Therefore, it is possible that certain regulatory elements are altered within the edited cell lines [20,21,22], which will ultimately result in differences in gene/protein expression.

Considering that CD44/CD44v overexpression has been described in CSCs [19,23,24,25,26,27], a subpopulation of tumor cells believed to be responsible for the initiation/maintenance of tumor growth and metastization [28,29,30], we also investigated whether CD44v6 affects one of the most important CSC feature, their self-renewal capacity. Interestingly, we show that either complete CD44v6 KD or specific exon-v6 skipping from GC cells leads to the severe loss of self-renewal capacity. Many cancer patients experience tumor recurrence because chemotherapy often fails to eliminate CSCs, which are frequently more chemoresistant than the remaining tumor bulk [23]. Being a marker of stemness, CD44v6 is associated with CSC features, like therapy resistance and metastization [19,23]. Therefore, we demonstrate that removing only exon-v6 results in CSC feature attenuation, by decreasing the self-renewal capacity and resistance to cisplatin of GC cells. In addition to v6 being relevant to drug resistance and cell self-renewal in gastric cancer cells, it is possible that other CD44v exons, such as v9, also play a role in these capabilities. For instance, CD44v9 has been described to confer drug resistance to gastric cancer cells [31,32], and we have observed that cells that lack exon v9 in addition to exon v6 (both MKN45 and GP202 CD44v6 KD cells) consistently present increased sensitivity to cisplatin and 5-FU, in the long term cell survival (clonogenic) assay. Therefore, in this context, it is possible that the v9 exon is also contributing (in addition to v6) to the drug resistance phenotype. To clarify this issue, which falls beyond the scope of the present study, similar experiments (to the ones we undertook here) could be performed, using CRISPR/Cas9 cells lacking specifically CD44v9.

Although we successfully used CRISPR/Cas9 to generate our study models, this strategy is far from being used as a therapeutic strategy, due to the technical and ethical issues that it raises [33,34]. Therefore, we tested another strategy to replicate the CRISPR/Cas9 results (albeit in a transient manner). We used PMOs, a strategy that has been implemented in the clinics. Indeed, PMOs are the only therapeutic option for Duchenne muscular dystrophy patients, where skipping the mutated exon can restore the reading frame, restoring protein production [35]. Hence, CRISPR/Cas9 and PMOs are good techniques for splicing modulation, being the former the best to create stable disease study models and the later the best to apply into pre-clinical studies. We strongly believe that using a PMO approach in CD44v6 expressing tumors would lead to tumor sensitization to chemotherapy and delayed tumor progression. Nevertheless, more studies using this approach, particularly using in vivo models, would be important to support our findings.

## 4. Materials and Methods

All reagents were purchased from ThermoFisher Scientific (Waltham, MA, USA), unless otherwise stated.

### 4.1. Cell Culture

Human GC cell line MKN45 was purchased from the Japanese Collection of Research Bioresources Cell Bank, and the non-commercial GP202 cell line was established at IPATIMUP [36]. Both cell lines were cultured in RPMI with 10% fetal bovine serum (FBS) (Biowest, Nuaillé, France), and maintained at 37 °C under 5% CO2 humidified atmosphere. Cells were grown in the absence of antibiotics (unless when puromycin selection was performed) and confirmed to be mycoplasma free.

### 4.2. Generation of a Permanent Exon-v6 Skipping Model by CRISPR/Cas9

CRISPR/Cas9 is a gene editing tool to permanently modify the genome, either by insertions, deletions or point mutations. sgRNAs were specifically designed to target a specific DNA portion via Watson-Crick base paring, and thus precisely guide the Cas9 protein to the region to be edited. The Cas9 protein recognizes a protospacer adjacent motif (PAM) sequence, causing a double-strand break (DSB) in the target DNA, that can then be repaired by the non-homologous end-joining (NHEJ) or the homology-directed repair (HDR) pathways [37]. All sgRNAs were designed to target the exon-v6 adjacent areas using Benchling online platform, and were purchased from Sigma-Aldrich (Poole, UK). Six sgRNAs were designed (Figure 1B,C) and individually cloned into pSpCas9(BB)-2A-Puro (PX459) V2.0 plasmid (Addgene plasmid #62988, Watertown, MA, USA), in the Bbs I restriction site, according to the “Morrisey Lab Protocol” [38], with few adaptations. Each Sg plasmid was transformed into Escherichia coli one shot Match1TM-T1R (Invitrogen, Carlsbad, CA, USA), according the manufacturer’s instructions. A colony PCR was performed to screen the colonies using the Primers A and D from Table 1. The positive colonies were amplified and sequenced using the same primers. The sgRNA were transfected in pairs to MKN45 and GP202 GC cell lines. As control, GC cell lines were transfected with an empty vector (mock). Briefly, cells were plated in 12 well plates and allowed to grow for 24 h, until approximately 70% confluency. Afterwards, Sgs’ plasmid or the empty plasmid were complexed with 1.75 µL of Lipofectamine 3000 transfection reagent, according to the manufacturer’s instructions. After 48 h, puromycin (Merck, Darmstadt, Germany) was added to the cells to select vector positive cells. Puromycin was renewed every 72 h, until the non-transfected cells, which do not have the antibiotic selection gene to puromycin carried by the plasmid, were dead.

### 4.3. Genotyping Analysis of CRISPR/Cas9 Skipping Models

GDNA was extracted using NZY Tissue gDNA Isolation kit (NZYTech, Lisbon, Portugal) and RNA was isolated using TriPure Isolation Reagent (Sigma-Aldrich, Poole, UK), according to the manufacturers’ protocol. NanoDrop^®^ ND-1000 UV-Vis Spectrophotometer (ThermoFisher Scientific, Waltham, MA, USA),was used to determine gDNA and RNA quality and concentration. cDNA was synthesized using 1 µg of template RNA and SuperScript^®^ II reverse transcriptase, according to the manufacturers’ protocol. Non-RT negative controls were produced replacing SuperScript^®^ II for sterile water. gDNA and cDNA were amplified with multiplex PCR kit (Qiagen, Venlo, The Netherlands) and primers designed to flank exon-v6, in the flanking introns (primers C/F were used to characterize gDNA) or flanking exons (primers B/E were used to characterize cDNA). PCR products were analyzed through 2% agarose gel electrophoresis and sanger sequencing.

### 4.4. CD44/CD44v Expression Analysis

CD44v6 and total CD44 gene expression were assessed by RT-qPCR, using specific probes for CD44v6 (exon span v5–v6, Hs.PT.58.45400024) and CD44 total (exon span 2–3, Hs.PT.58.4880087) (both from iDT, Coralville, IA, USA). RT-qPCR was performed in triplicate, using 1 µL of template cDNA per well. TaqMan Master Mix was used to amplify PCR product and ABI Prism 7500 Fast Sequence Detection System was used to quantify the expression. Relative expression was calculated by comparative 2^−ΔΔCт^ method using the housekeeping gene 18 s (custom assay) (iDT, Coralville, IA, USA).

CD44v6 and total CD44 protein expression were assessed by flow cytometry. Cells were detached using versene reagent and blocked with 3% bovine serum albumin (BSA)–phosphatase buffer saline (PBS), for 30 min. Cells were incubated with the following primary antibodies: mouse monoclonal antibody against total CD44 (156-3C11; 1:100 dilution; 60 min; Cell Signaling Technology, Beverly, MA, USA) and CD44v6 (MA54; 1:100; 60 min). Cells were washed and subsequently incubated with secondary antibody anti-mouse Alexa Fluor 647 (1:500; 30 min). Fluorescence was measured using FACS Canto II (BD Biosciences, Franklin Lakes, NJ, USA). Flow Jo version 10 software was used to analyze the data.

Total CD44, CD44v6 and CD44v9 protein expression levels and subcellular location were assessed by immunofluorescence. After cell growth in glass coverslips, cells were fixed with 4% paraformaldehyde (PFA) (Merck, Darmstadt, Germany). Cells were permeabilized using 0.2% Triton X-100 (Sigma-Aldrich) and subsequently blocked with 5% BSA-PBS. The primary antibodies used were mouse monoclonal antibody against CD44v6 (MA54; 1:100; overnight (ON); 4 °C), rat monoclonal antibody against CD44v9 (RV3; 3 µg/mL; ON; 4 °C; AB Nova, Taipé, Taiwan), and mouse monoclonal antibody against CD44 total (156-3C11; ON; 1:100 dilution; Cell Signaling Technology, Beverly, MA, USA). After washings with PBS, cells were incubated with the appropriate secondary antibodies: Alexa Fluor 594 Goat Anti-Mouse secondary antibody (1:500; 2 h; Life Technologies, Carlsbad, CA, USA) and Alexa Fluor 488 Goat Anti-Rat secondary antibody (1:500; 2 h; Life Technologies). Nuclei were stained with DAPI (1:1000; 5 min; Sigma-Aldrich). Vectashield mounting media (Vector Laboratories, Burlingame, CA, USA) was used to mount the cover slips. Fluorescence was measured (AxioCam fluorescence microscope, Zeiss, Gottingen, Germany) and data were analyzed using AxioVision software version 4.8. (Rockville, MD, USA).

### 4.5. Inhibition of CD44v6 Expression by siRNA

Mock cells and CRISPR/Cas9 edited cells were seeded in 6 well plates (1.5 × 10^5^ and 2 × 10^5^ cells/well for MKN45 and GP202, respectively) and allowed to adhere ON. In the following day, Mock cells were transfected with 20 nM scramble (negative control; DS NC1; iDT, Coralville, IA, USA) or CD44v6 siRNA (Sense strand: 5′-GCGUCAGGUUCCAUAGGAAUCCUTT-3′; Antisense strand: 5′-AAAGGAUUCCUAUGGAACCUGACGCAG-3′; iDT), using Lipofectamine^®^ RNAiMax transfection reagent, according to the manufacturers’ instructions. CRISPR/Cas9 edited cells were also transfected with scramble siRNA to ensure the same conditions between mock, KD and CD44v6 edited cells for the following assays. After 24 h of incubation, transfected cells were detached, counted and re-plated in 96-(2500 and 4000 cells for MKN45 and GP202 respectively) and 6-well plates (1000 and 3000 cells for MKN45 and GP202 respectively), for short-and long-term assays, respectively.

### 4.6. Cell Survival Assays

After transfection, and 24 h after cell re-plating, cells were treated with cisplatin (reconstituted in 0.9% NaCl), 5-FU (reconstituted in sterile water) or vehicle (0.9% NaCl). The concentrations used in short-term assays were: 2.5 µM (MKN45) and 20 µM (GP202) for cisplatin and 5 µM (MKN45 and GP202) for 5-FU. The short-term effects of cisplatin and 5-FU on cell survival were assessed after 48 h of treatment, using PB (Invitrogen) and SRB (Sigma-Aldrich) assays. PB reagent was diluted in RPMI to 1x and added to cells. Cells were incubated for 45 min at 37 °C and fluorescence measured (with an excitation wavelength of 560 nm and emission of 590 nm). Afterwards, cells were fixed in 10% trichloroacetic acid (TCA) (Merck, Darmstadt, Germany) for 1 h on ice, and proteins were stained with 4% SRB solution (Sigma-Aldrich), for 30 min at room temperature. Wells were repeatedly washed with 1% acetic acid to remove unbound dye. Protein stain was solubilized with 10 mM Tris solution (Calbiochem, San Diego, CA, USA) and absorbance read at 560 nm, with background correction at 655 nm. Fluorescence and absorbance were assessed in a microplate reader PowerWave HT Microplate Spectrophotometer (BioTek, Bad Friedrichshall, Germany). To calculate the percentage of cell survival, fluorescence (Fluo)/absorbance (Abs) readings of treatment conditions were normalized to the vehicle (% cell survival = (Treatment _Fluo/Abs_ × 100)/Vehicle _Fluo/Abs_).

The long-term effects of cisplatin and 5-FU treatment were analyzed using the clonogenic assay. Briefly, after transfection and 24 h after cell re-plating, cells were treated with 0.5 µM cisplatin, 2.5 µM or 2 µM of 5-FU (for MKN45 and GP202, respectively) and vehicle. After 48 h drug treatment, RPMI medium was renewed, and cells were incubated under normal conditions (5% CO_2_ humidified atmosphere at 37 °C) for ~10 days, until visible colonies had about 50 cells each. Afterwards, cells were fixed with methanol (Fisher Scientific, Hampton, NH, USA), stained with 0.5% crystal violet (Sigma-Aldrich), and colonies counted. To calculate the percentage of cell survival, the number of colonies of each treatment conditions was normalized to vehicle (% cell survival = (Treatment_No. of colonies_ × 100)/Vehicle_No. of colonies_).

### 4.7. Transient Exon-v6 Skipping Using Morpholinos

To understand the therapeutic potential of exon-v6 skipping, we tested a transient strategy approved in the clinics for other diseases. To this end, we checked the possible splice sites near exon-v6, using Human Splicing Finder online tool [39], and designed PMOs to target the specific splice acceptor and donor at the 5′ and 3′ sites of exon-v6, respectively. Two PMOs were designed: “CD44v6_BEG” (5′-CTGGACTGTGAGAAGAATATCAGTT-3′), at the 5′ site of exon-v6 and “CD44v6_END” (5′-CTTGTTAAACCATCCATTACCAGCT-3′), at the 3′ site of exon-v6. Both GC cell lines were seeded in 12-well plates and allowed to adhere ON, until approximately 60–70% confluency. In the following day, culture medium was replaced and PMOs (alone or mixed) were transfected into cells using Endo-Porter transfection reagent (Gene Tools, Philomath, OR, USA) and incubated for 48 h. The concentrations of each PMO used to transfect cells were 4 µM along with 2 µM of Endo–Porter. After this period, cells were collected, either for genotyping or immunofluorescence analysis.

### 4.8. Statistical Analysis

Statistical analysis was performed using GraphPad Prism version 7.00 software (GraphPad Software Inc., San Diego, CA, USA). A normality test was performed using a Shapiro-Wilk test. Two-way ANOVA with Tukey’s post hoc test for multiple comparison analysis was used, with 95% confidence interval. Significant differences were considered significant when *p* < 0.05.

## 5. Conclusions

In summary, we demonstrated that depletion of exon-v6 from GC cells can increase sensitivity to cisplatin and impair self-renewal, which could prove to be an extremely valuable approach to improve GC patient therapy and survival. Moreover, we are confident that a PMO approach triggering exon-v6 skipping may be a feasible therapeutic option to sensitize tumor cells and possibly condition the progression of CD44v6 positive gastric cancers.

## Figures and Tables

**Figure 1 cancers-12-02378-f001:**
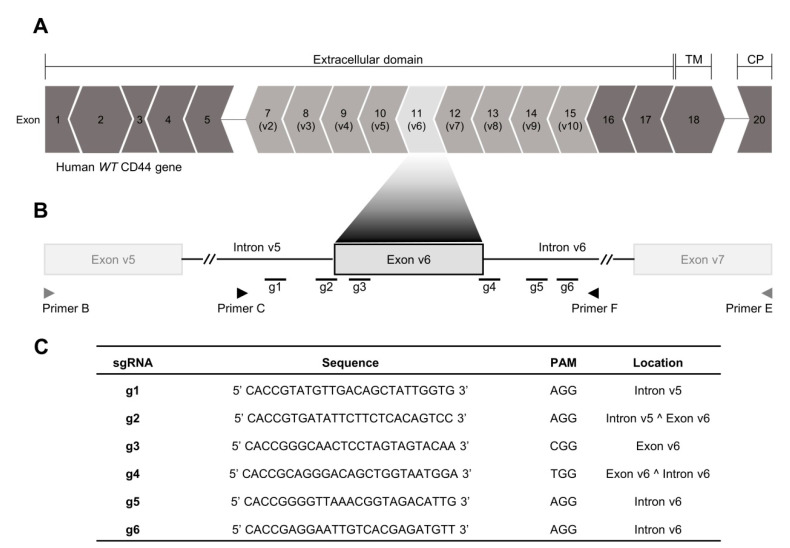
Human CD44 locus and CRISPR/Cas9 single guide RNA (sgRNA) design strategy. (**A**) The human CD44 canonical transcripts annotated with 18 exons (ENST00000428726.8). The dark grey exons comprise the constitutively expressed portion of the gene (exons 1–5, 16–18 and 20), and the light grey corresponds to the alternatively expressed portion of the gene in humans (exons v2–v10). Each exon is docked to adjacent exons in the mRNA sequence. Every set of three nucleotides encodes the amino acids that will comprise the CD44 protein. In the exon boundaries, the resultant amino acid may enclose the last nucleotide of the former exon and the two first nucleotides from the next exon (>), or contain the two last nucleotides of the former exon and the first nucleotide from the next exon (<). This indicates that if exon-v6 is removed, exons v5 and v7 are spliced in-frame. (**B**) Genomic spots of the sgRNA designed to target the adjacent areas to exon-v6. Primers B and E (represented in grey) were used to genotype the complementary DNA (cDNA) and primer C and F (represented in black) were used to genotype the genomic DNA (gDNA) of the edited cells. (**C**) Specifications of the sgRNAs (g1–g6) designed near the exon-v6 vicinity to permanently induce exon-v6 skipping. TM—Transmembrane domain; CP—Cytoplasmic domain; WT—wild-type; PAM—Protospacer Adjacent Motif.

**Figure 2 cancers-12-02378-f002:**
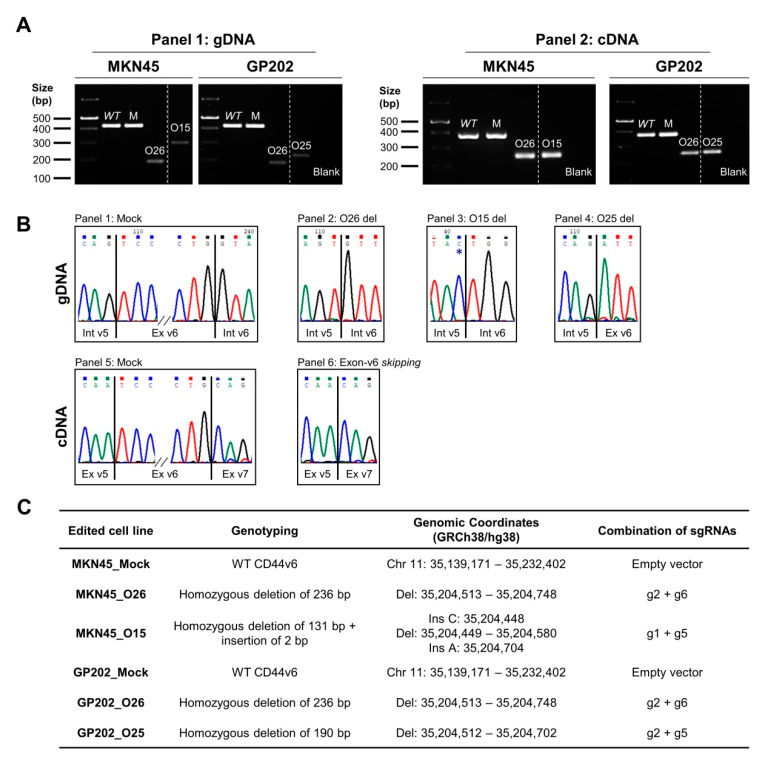
Genotyping analysis of CRISPR/Cas9 edited GC cells. (**A**) Analysis of gDNA (Panel 1) and cDNA (Panel 2) from wild-type (WT), Mock and edited GC cell lines. In Panel 1, MKN45/GP202_Mock cells present a transcript of the same size of WT (422 bp), while the edited cell lines present a transcript with smaller sizes (MKN45_O26 and GP202_O26–186 bp; MKN45_O15–293 bp; GP202_O25–232 bp). In Panel 2, MKN45/GP202_WT and Mock cells present a transcript with a size of 378 bp, while all edited cell lines present a smaller transcript (249 bp) corresponding to the occurrence of exon-v6 skipping. All PCR products were run in the same 2% agarose gel (Appendix A). (**B**) Representative examples of sequencing analysis: Panel 1—gDNA of Mock cells; Panel 2—gDNA deletion of 235 bp of O26 clones. (The present sequencing analysis was performed with gDNA extracted from MKN45_O26, but GP202_O26 presents the same deletion); Panel 3—gDNA deletion of 131 bp of O15 clone (* C nucleotide is a single nucleotide insertion); Panel 4—gDNA deletion of 190 bp of O25 clone; Panel 5—cDNA of Mock cells; Panel 6—cDNA of all edited clones, which represents the skipping of exon-v6. (**C**) Specifications of each edited cell line by CRISPR/Cas9 with the respective edition genomic coordinates (GRCh38/hg38) and the combination of sgRNAs transfected to create each edited cell line. WT–wild-type; M–Mock; gDNA–genomic DNA; cDNA–complementary DNA; Chr—Chromosome; Del—Deletion; Ins—Insertion.

**Figure 3 cancers-12-02378-f003:**
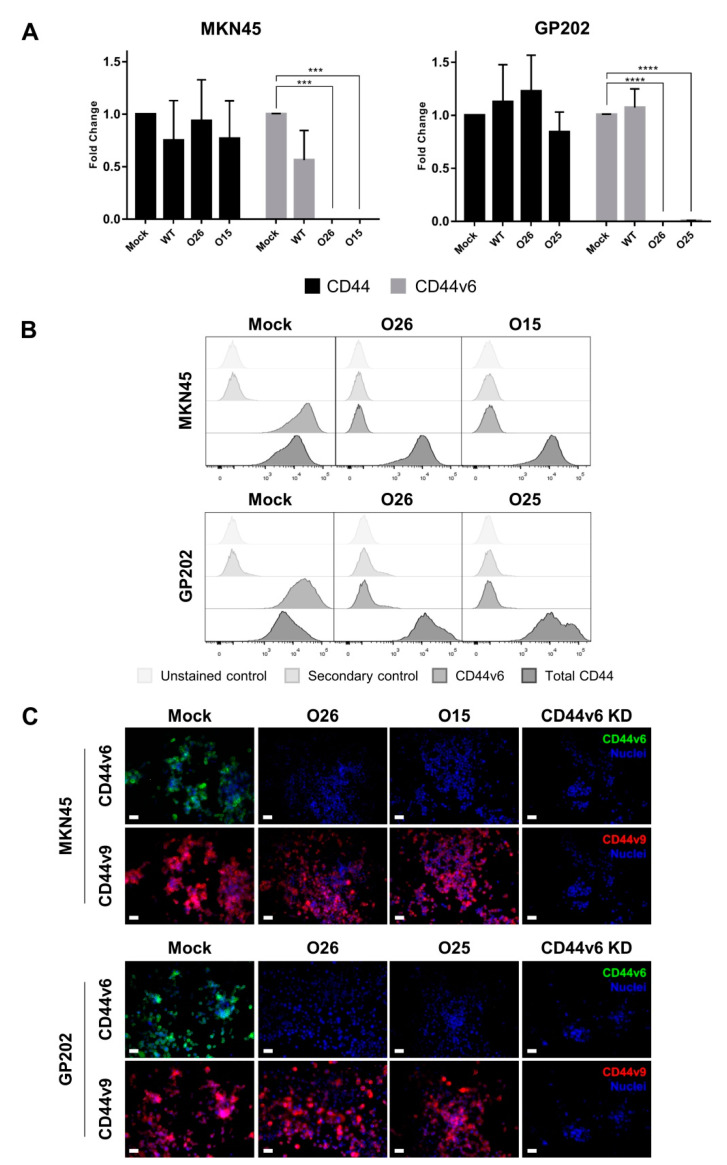
Characterization of CRISPR/Cas9 edited cells and CD44v6 knockdown (KD) cells (using siRNA) at the RNA and protein level for two GC cell lines. (**A**) RT-qPCR representing the fold-change of CD44v6 and total CD44 gene expression in MKN45 and GP202 WT, Mock and edited cells. Results represent the mean + SD of three independent experiments. Statistically significant results were determined by Two-way ANOVA with Tukey’s Post Hoc Test for multiple comparison analysis (*** *p* < 0.001; **** *p* < 0.0001). (**B**) Protein expression analysis by flow cytometry for CD44v6 and total CD44 expression in GP202 and MKN45 cells; (**C**) Co-immunofluorescence analysis for CD44v6 (green) and CD44v9 (red) expression in GP202 and MKN45 cells. To facilitate the visualization, CD44v6 and CD44v9 panels are depicted in separate, but were performed in the same slide, and membranous CD44v6 and CD44v9 staining are shown in greater detail in Appendix A. Nuclei were stained with DAPI (blue) and white scale bars represent a distance of 50 µm. WT—wild-type; KD—Knockdown.

**Figure 4 cancers-12-02378-f004:**
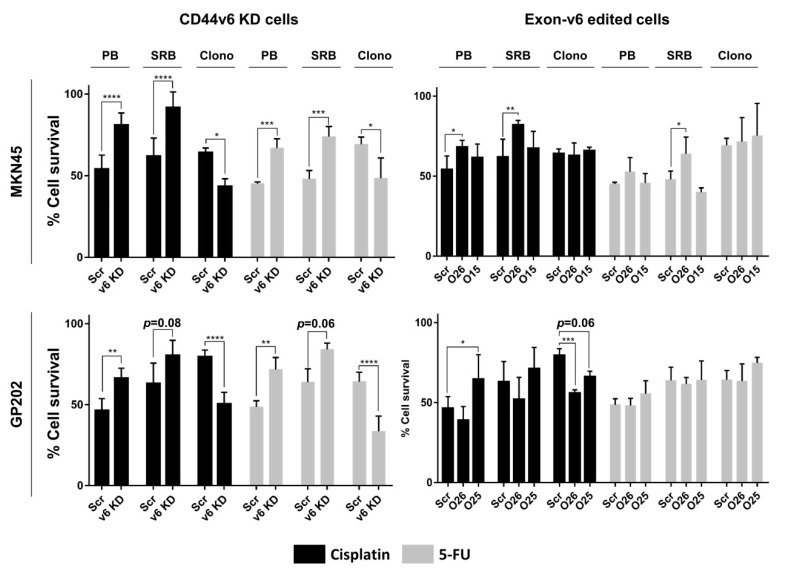
Cell survival in response to cisplatin or 5-fluorouracil (5-FU) in MKN45 and GP202 cells. Percentage cell survival of CD44v6 expressing cells (Scramble) vs. CD44v6 KD cells and Scramble vs. exon-v6 depleted cells (O26, O15, O25) to cisplatin (represented in black) and 5-FU (represented in grey) treatment analyzed by Presto Blue (PB) and Sulforhodamine B (SRB) assays, after 48 h and clonogenic assay, after 10 days. The concentrations used in PB and SRB assays were: 2.5 µM (MKN45) and 20 µM (GP202) of cisplatin and 5 µM (MKN45 and GP202) of 5-FU. The concentrations used for the clonogenic assay were: 0.5 µM (MKN45 and GP202) for cisplatin and 2.5 µM (MKN45) and 2 µM (GP202) for 5-FU. Results represent the mean + SD of three independent experiments. Statistically significant results were determined by two-way ANOVA with Tukey’s Post Hoc Test for multiple comparison analysis (* *p* < 0.05; ** *p* < 0.01; *** *p* < 0.001; **** *p* < 0.0001). Src–Scramble; v6 KD–CD44v6 Knockdown; PB–PrestoBlue; SRB–Sulforhodamine B; Clono–Clonogenic; KD–Knockdown.

**Figure 5 cancers-12-02378-f005:**
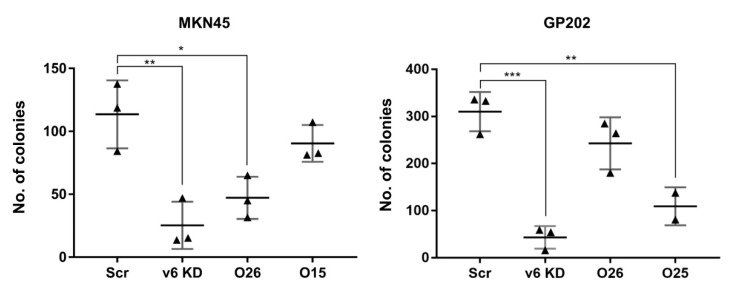
Self-renewal capacity of MKN45 and GP202 cells in response to the KD of CD44v6 isoforms or specific exon-v6 skipping. Colony formation capacity of Scramble, CD44v6 KD cells and exon-v6 edited cells lines. Results show the mean ± SD of three independent experiments. Statistically significant results were determined by two-way ANOVA with Tukey’s Post Hoc Test for multiple comparison analysis (* *p* < 0.05; ** *p* < 0.01; *** *p* < 0.001). Src–Scramble; v6 KD–CD44v6 Knockdown; KD—Knockdown.

**Figure 6 cancers-12-02378-f006:**
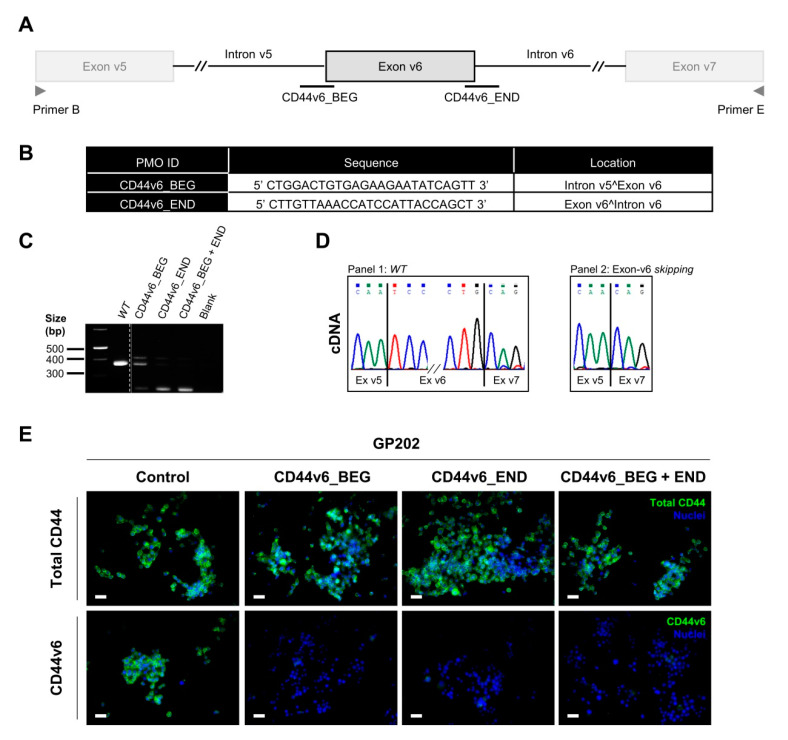
Characterization of transfected cells established from WT GP202 GC cell line, using a PMOs approach. (**A**) Scheme of the PMOs designed to target the splice-sites of exon-v6. Primers B and E (represented in grey) were used to genotype the cDNA of cells. (**B**) Specifications of PMOs (CD44v6_BEG and CD44v6_END) designed to target the splice-sites of exon-v6 and cause exon-v6 skipping. (**C**) Analysis of cDNA from WT and transfected cells with either single or mixed PMOs. Transfected cells present a smaller transcript (249 bp) than WT cells (378 bp), corresponding to the skipping of exon-v6. All PCR products were analyzed in the same 2% agarose gel (Appendix A). (**D**) Representative examples of the sequencing analysis: Panel 1—WT 378 bp transcript, which corresponds to the WT cDNA sequence; Panel 2—249 bp transcript of the transfected cells, which corresponds to the skipping of exon-v6. The sequencing data was generated from cDNA extracted from CD44v6_END transfected cells (**E**) Immunofluorescence analysis depicting total CD44 (green) and CD44v6 expression (green) of GP202 WT and transfected cells. To facilitate visualization, membranous CD44v6 staining is shown in greater detail in Appendix A. Nuclei were stained with DAPI (blue) and white scale bars represent a distance of 50 µm. WT—wild-type; cDNA—complementary DNA.

**Table 1 cancers-12-02378-t001:** Details of the primers designed to validate CRISPR/Cas9 constructs (primer A and D) and genotype the exon-v6 skipping cell models obtained by CRISPR/Cas9 and Phosphorodiamidate Morpholino Oligomers (PMOs) approaches (primer B, C, E and F).

Orientation	In-Text Name	Binding Site	Primer Sequence (5′–3′)	Melting Temperature
Forward	A	pSpCas9(BB)_Bbs I	5′-GGGCCTATTTCCCATGATTCCTT-3′	Tm = 68.5 °C
	B	CD44 Exon 10 (v5)	5′-ATGTAGACAGAAATGGCACCAC-3′	Tm = 62.7 °C
	C	CD44 Intron 10	5′-ATCAGTGGCCTGTTTCCTTG-3′	Tm = 64.0 °C
Reverse	D	pSpCas9(BB)_Bbs I	5′-GACTCGGTGCCACTTTTTCAA-3′	Tm = 66.2 °C
	E	CD44 Exon 12 (v7)	5′-CCATCCTTCTTCCTGCTTGATG-3′	Tm = 66.8 °C
	F	CD44 Intron 11	5′-TTTGGCTCTGTGTGAACTGC-3′	Tm = 64.1 °C

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
