# Peer review of "Skipping Exon-v6 from CD44v6-Containing Isoforms Influences Chemotherapy Response and Self-Renewal Capacity of Gastric Cancer Cells"

_cancers, 2020, doi:10.3390/cancers12092378_

Round 1

Reviewer 1 Report

This study addressed the exon-v6 of CD44v6 isoforms in gastric cancer patients. I have one suggestion about this study. 

In terms of this phenomenon, is there any difference between western and western populations?

Author Response

Thank you for your comment which we have addressed below. At the end of the document, we also present a list of small changes that were introduced to the manuscript. We hope you find that we addressed your comment in a suitable manner and that this new version of the manuscript meets the expectations of the Reviewers and Editors and fulfills the criteria to be accepted for publication in Cancers.

Point 1: In terms of this phenomenon, is there any difference between western and western populations?

Response to point 1: Thank you for your comment. We understood you intended to question if there are any differences between Western and Eastern populations regarding the type and expression of CD44v6-containing isoforms in gastric cancer.

CD44v6-containing isoforms encompass a group of isoforms including variant exon-6 (v6). These isoforms are generated by the introduction of alternatively spliced exons, therefore the diversity and relative RNA expression of different CD44v6-containing isoforms in gastric cancer may differ between patients and between different populations. To the best of our knowledge, there are no reports in the literature that have specifically addressed this subject. Regarding the protein expression levels of CD44v6-containing isoforms in Western and Eastern populations, to the best of our knowledge there is at least one meta-analysis report (Li et al., 2016) that used immunohistochemistry. This study included tumors from patients from different countries, including Eastern and Western countries. In this report it becomes clear that the percentage of gastric tumors with positive CD44v6 varies between studies, and consequently between different populations, but it is not clear whether countries from different world regions have differences in CD44v6 expression, since expression also varies widely between different studies from the same country.

As this remains a vastly unknown subject, we opted for not adding this information to the manuscript text.

Reference:

Lu, L.; Huang, F.; Zhao, Z.; Li, C.; Liu, T.; Li, W.; Fu, W. CD44v6: a metastasis-associated biomarker in patients with gastric cancer?: a comprehensive meta-analysis with heterogeneity analysis. Medicine. 2016, 95, e5603, DOI: 10.1097/MD.0000000000005603

Additional modifications to the original text:

  • In Fig. 2C, we added an “_” to “MKN45 Mock” and “GP202 Mock” to maintain the consistency between the names of edited cell lines.
  • In Fig. 2 caption, lines 120 and 122, we added “MKN45/GP202_”.
  • In both graphs of Fig. 5, we corrected the abbreviation of “number” from “Nº” to “No.” in the y-axis.
  • In the materials and methods section 4.6 in line 404, we corrected the abbreviation of “number” from “Nº” to “No.”.
  • In supplementary materials section, lines 432-435, we added the description of two additional supplementary figures (Figure S2 and Figure S4) as follows: “Figure S2: Illustrative immunofluorescence images highlighting CD44v6 and CD44v9 membranous staining observed in Figure 3C. (…) Figure S4: Illustrative immunofluorescence images highlighting CD44v6 membranous staining observed in Figure 6E.”
  • The numbering of the supplementary Figures had to be changed and the Figure S.2 in the previous manuscript version is now Figure S3.
  • For consistency, we added a dot to the authors’ initials in lines 449, 450, 451, 457 and 459.
  • In references section, lines 541-547, we added two references (number 31 and 32) and re-numbered the references onwards.
  • The resolution of Main Figures 1 to 6 was improved and small formatting alterations were introduced in Figures 2 to 6. Therefore, all main Figures were substituted by new ones in the word document and a Zip file containing the high-resolution Figures is also being uploaded.

Reviewer 2 Report

In this manuscript, the author demonstrated that exon-v6 specific removal from CD44v isoforms using by CRISPR/Cas9 systems increases cell sensitivity to cisplatin and impairs GC cells' self-renewal. This is a very interesting study as a therapeutic strategy targeting the variant isoform of CD44 using CRISPR/Cas9 systems.

(1) In CD44 variant isoforms, CD44v9 has been well known to be associated with the development of drug (5-Fluorouracil, or cisplatin) resistance, and further the development of gastric cancer. In this study, in CD44v6-KD cells, both of CD44v6 and CD44v9 was disappeared.  It was unclear which of CD44v6 and CD44v9 was more strongly associated with drug resistance or gastric carcinogenesis. The author better to check the drug resistance in exon-v9 specific removal cells using by CRISPR/Cas9, and to compare to that in exon-v6 specific removal cells.

2) CD44 is a cell surface protein, and therefore if you analyze the immunostaining for CD44variant, staining signals are observed in the extracellular membrane. In Figure 3C and 6E, these staining signals are unclear and therefore it can not be confirmed that these signals are CD44 protein. If immunostaining analysis is too difficult for you, you can effectively use the Flow cytometric analysis for CD44 protein.

Author Response

Thank you for your comments. We have addressed each one of them point-by-point by adding the requested information in the relevant sections of this manuscript. At the end of the document, we also present a list of small changes that were introduced to the manuscript. All alterations to the original manuscript are shown in “track changes” and highlighted in yellow. We hope you find that we addressed your comments in a suitable manner and that this new version of the manuscript meets the expectations of the Reviewers and Editors and fulfills the criteria to be accepted for publication in Cancers.

Point 1: In CD44 variant isoforms, CD44v9 has been well known to be associated with the development of drug (5-Fluorouracil, or cisplatin) resistance, and further the development of gastric cancer. In this study, in CD44v6-KD cells, both of CD44v6 and CD44v9 was disappeared.  It was unclear which of CD44v6 and CD44v9 was more strongly associated with drug resistance or gastric carcinogenesis. The author better to check the drug resistance in exon-v9 specific removal cells using by CRISPR/Cas9, and to compare to that in exon-v6 specific removal cells.

Response to point 1: Thank you for your suggestion, which we consider very pertinent.

Indeed, CD44v9 has been associated with 5-FU and cisplatin resistance. In our CD44v6-KD cell model, transcripts encompassing simultaneously v6 and v9 exons were downregulated. In this context, it is not possible to know which of these exons is more strongly associated with the drug resistance we observed. Nevertheless, when we delete specifically exon v6 (by CRISPR/Cas9) from a CD44v6-containing isoform that still maintains the v9 exon, resistance to cisplatin decreases in two of the edited GC cells lines, and so does their self-renewal capacity (Fig. 4). This indicates that exon v6, by itself, is capable of modulating drug resistance and self-renewal capacity in gastric cancer cells. Indeed, in those CRISPR/Cas9 edited cells that lack exon v6 and retain exon-v9, the presence of v9 is not able to rescue the drug resistance and self-renewal capacities.

In this study, the analysis of exon-v9 expression was intended to complement the characterization of the CRISPR/Cas9 edited cell lines. We proved, by flow cytometry, that our exon-v6 skipping cell lines have no CD44v6 expression while preserving total CD44 expression (Fig. 3B). On the other hand, by using an exon-v6 directed RNAi-based approach, we demonstrate that by targeting exon v6, all v6-containg isoforms were destroyed, including those containing exon v9. This strongly supports that v6 and v9 exons are both part of the same isoforms.

In contrast, the CRISPR/Cas9 exon-v6 skipping cells, present CD44v transcripts that only lack exon v6 while maintaining the expression of other portions of the protein (e.g. the v9 domain) (Fig. 3C).

The role of CD44v9 in drug resistance in gastric cancer cells was not within the scope of the present study. Indeed, we cannot rule out that exon v9 may also contribute (in addition to v6) to the drug resistance we identified in the CD44v6-KD cells. To assess this, further experiments would have to be performed (as mentioned by the reviewer). To highlight the importance of this subject, we included the following sentence in the discussion section, lines 272-281: “In addition to v6 being relevant to drug resistance and cell self-renewal in gastric cancer cells, it is possible that other CD44v exons, such as v9, also play a role in these capabilities. For instance, CD44v9 has been described to confer drug resistance to gastric cancer cells (Miyoshi et al., 2018; Mashima et al., 2019) and we have observed that cells that lack exon v9 in addition to exon v6 (both MKN45 and GP202 CD44v6 KD cells) consistently present increased sensitivity to cisplatin and 5-FU, in the long term cell survival (clonogenic) assay. Therefore, in this context, it is possible that the v9 exon is also contributing (in addition to v6) to the drug resistance phenotype. To clarify this issue, which falls beyond the scope of the present study, similar experiments (to the ones we undertook here) could be performed using CRISPR/Cas9 cells lacking specifically CD44v9.”

References:

Miyoshi, S.; Tsugawa, H.; Matsuzaki, J.; Hirata, K.; Mori, H.; Saya, H.; Kanai, T.; Suzuki, H. Inhibiting xCT improves 5-fluorouracil resistance of gastric cancer induced by CD44 variant 9 expression. Anticancer Res. 2018, 38, 6163-6170, DOI: 10.21873/anticanres.12969

Mashima, T.; Iwasaki, R.; Kawata, N.; Kawakami, R.; Kumagai, K.; Migita, T.; Sano, T.; Yamaguchi, K.; Seimiya, H. In silico chemical screening identifies epidermal growth factor receptor as a therapeutic target of drug-tolerant CD44v9-positive gastric cancer cells. Br J Cancer. 2019, 121, 846-856, DOI: 10.1038/s41416-019-0600-9

Point 2: CD44 is a cell surface protein, and therefore if you analyze the immunostaining for CD44variant, staining signals are observed in the extracellular membrane. In Figure 3C and 6E, these staining signals are unclear and therefore it can not be confirmed that these signals are CD44 protein. If immunostaining analysis is too difficult for you, you can effectively use the Flow cytometric analysis for CD44 protein.

Response to point 2: Thank you for your comment. It is true that in some images, displayed in the Manuscript file, the CD44 membrane staining is obscured by the image background, hampering its clear identification at the cell membrane. This occurs due to the decrease of resolution when the image is inserted in the Word file. To try to avoid this, we have improved further the quality of these images and provided an additional supplementary figure (Figure S2) highlighting the membranous staining of CD44v6 and CD44v9 observed in Fig. 3C. Moreover, we would like to invite the reviewer to analyze the high-resolution files that we provided in a separate file upon submission.

We have also prepared a new supplementary figure (Fig.S2) and included the following sentence in lines 144 and 145 (Legend of Figure 3) to allow clear visualization of the membranous staining of CD44v6 and CD44v9: “…and membranous CD44v6 and CD44v9 staining are shown in greater detail in Fig.S2.”

We are confident that these improved images will convince the Reviewer that not additional experiments are needed. When characterizing the CRISPR/Cas9 edited cell lines, we confirmed the expression of total CD44 and lack of CD44v6 by flow cytometry (Fig. 3B). This supports our confidence in the CD44v6 immunostaining results presented in Fig. 3C.

Although we have not characterized the CD44/CD44v6 expression of gastric cancer cells by flow cytometry upon v6-skipping using PMO’s approach (Fig. 6E), we have now also improved the quality of images and included an additional supplementary figure (Figure S4) highlighting the membranous staining of CD44v6 observed in Fig. 6E. Therefore, we have included the following sentence in line 228 and 229 (Legend of Figure 6): “To facilitate visualization, membranous CD44v6 staining is shown in greater detail in Fig.S4.”

We did not perform similar flow cytometry experiments for CD44v9, as we used v9 staining only to demonstrate that both exons v6 and v9 were part of the same transcript, and only v6 was removed upon specific CRISPR/Cas9 editing. We hope the Reviewer understands and accepts that studying further the specific role of v9 exon is beyond the scope of this manuscript.

Additional modifications to the original text:

  • In Fig. 2C, we added an “_” to “MKN45 Mock” and “GP202 Mock” to maintain the consistency between the names of edited cell lines.
  • In Fig. 2 caption, lines 120 and 122, we added “MKN45/GP202_”.
  • In both graphs of Fig. 5, we corrected the abbreviation of “number” from “Nº” to “No.” in the y-axis.
  • In the materials and methods section 4.6 in line 404, we corrected the abbreviation of “number” from “Nº” to “No.”.
  • In supplementary materials section, lines 432-435, we added the description of two additional supplementary figures (Figure S2 and Figure S4) as follows: “Figure S2: Illustrative immunofluorescence images highlighting CD44v6 and CD44v9 membranous staining observed in Figure 3C. (…) Figure S4: Illustrative immunofluorescence images highlighting CD44v6 membranous staining observed in Figure 6E.”
  • The numbering of the supplementary Figures had to be changed and the Figure S.2 in the previous manuscript version is now Figure S3.
  • For consistency, we added a dot to the authors’ initials in lines 449, 450, 451, 457 and 459.
  • In references section, lines 541-547, we added two references (number 31 and 32) and re-numbered the references onwards.
  • The resolution of Main Figures 1 to 6 was improved and small formatting alterations were introduced in Figures 2 to 6. Therefore, all main Figures were substituted by new ones in the word document and a Zip file containing the high-resolution Figures is also being uploaded.

Reviewer 3 Report

This is a very interesting and informative piece of work. I have very few concerns about this work.

  1. Fig4. The colors in bar graphs are not so obvious. The x-axis legends can be abbreviated.
  2. Fig3. High resolution picture will be preferred
  3. More cell lines preferably needs to be investigated in future to increase number of biological replicates.

Author Response

Thank you for your comments. We have addressed each one of them point-by-point by adding the requested information in the relevant sections of this manuscript. At the end of the document, we also present a list of small changes that were introduced to the manuscript. All alterations to the original manuscript are shown in “track changes” and highlighted in yellow. We hope you find that we addressed your comments in a suitable manner and that this new version of the manuscript meets the expectations of the Reviewers and Editors and fulfills the criteria to be accepted for publication in Cancers.

Point 1: Fig4. The colors in bar graphs are not so obvious. The x-axis legends can be abbreviated.

Response to point 1: Thank you for your comment. We have changed the colors of the bar graphs in Fig.4 to a lighter grey to facilitate the differentiation between groups. Also, we abbreviated “Scramble” and “CD44v6 KD” designations to “Scr” and “v6 KD”, respectively, on the x-axis of those graphs. For consistency, we also abbreviated these designations in Figure 5. In Fig. 4 and Fig. 5 caption, lines 177-178 and lines 199-200 respectively, we added the description of both abbreviations as follows: “Src – Scramble; v6 KD – CD44v6 Knockdown”.

Point 2: Fig3. High resolution picture will be preferred

Response to point 2: Thank you for your comment. We have now improved Fig. 3C and inserted it in the manuscript, below line 134. Notwithstanding, we also provided a zip archive with high resolution images.

Point 3: More cell lines preferably needs to be investigated in future to increase number of biological replicates.

Response to point 3: Thank you for your comment. We performed this study using two different gastric cancer cell lines in order to obtain cell line independent results. To increase the value of the study, we also used more than one exon-v6 skipping clone per cell line to avoid possible confounding results due to off-target effects. In a follow-up study, it would be important to increase the number of different cell lines. Taking into account this comment, we have included a mention to this in the Results section 2.3. in lines 193-194: “(…) however, in the future, it would be important to extend this analysis to additional GC cell lines.”.

Additional modifications to the original text:

  • In Fig. 2C, we added an “_” to “MKN45 Mock” and “GP202 Mock” to maintain the consistency between the names of edited cell lines.
  • In Fig. 2 caption, lines 120 and 122, we added “MKN45/GP202_”.
  • In both graphs of Fig. 5, we corrected the abbreviation of “number” from “Nº” to “No.” in the y-axis.
  • In the materials and methods section 4.6 in line 404, we corrected the abbreviation of “number” from “Nº” to “No.”.
  • In supplementary materials section, lines 432-435, we added the description of two additional supplementary figures (Figure S2 and Figure S4) as follows: “Figure S2: Illustrative immunofluorescence images highlighting CD44v6 and CD44v9 membranous staining observed in Figure 3C. (…) Figure S4: Illustrative immunofluorescence images highlighting CD44v6 membranous staining observed in Figure 6E.”
  • The numbering of the supplementary Figures had to be changed and the Figure S.2 in the previous manuscript version is now Figure S3.
  • For consistency, we added a dot to the authors’ initials in lines 449, 450, 451, 457 and 459.
  • In references section, lines 541-547, we added two references (number 31 and 32) and re-numbered the references onwards.
  • The resolution of Main Figures 1 to 6 was improved and small formatting alterations were introduced in Figures 2 to 6. Therefore, all main Figures were substituted by new ones in the word document and a Zip file containing the high-resolution Figures is also being uploaded.

Round 2

Reviewer 2 Report

The authors thoughtfully revised their manuscript.
Nothing further comments.